# Research on a Visual/Ultra-Wideband Tightly Coupled Fusion Localization Algorithm

**DOI:** 10.3390/s24051710

**Published:** 2024-03-06

**Authors:** Pin Jiang, Chen Hu, Tingting Wang, Ke Lv, Tingfeng Guo, Jinxuan Jiang, Wenwu Hu

**Affiliations:** College of Mechanical and Electrical Engineering, Hunan Agricultural University, Changsha 410128, China; 1233032@hunau.edu.cn (P.J.); huchen@stu.hunau.edu.cn (C.H.); 2711451490@stu.hunau.edu.cn (T.W.); lks@stu.hunau.edu.cn (K.L.); guotingfeng@stu.hunau.edu.cn (T.G.); jiangjinxuan@stu.hunau.edu.cn (J.J.)

**Keywords:** extended Kalman filtering, multi-sensor fusion, visual SLAM, ultra-wideband positioning

## Abstract

In the autonomous navigation of mobile robots, precise positioning is crucial. In forest environments with weak satellite signals or in sites disturbed by complex environments, satellite positioning accuracy has difficulty in meeting the requirements of autonomous navigation positioning accuracy for robots. This article proposes a vision SLAM/UWB tightly coupled localization method and designs a UWB non-line-of-sight error identification method using the displacement increment of the visual odometer. It utilizes the displacement increment of visual output and UWB ranging information as measurement values and applies the extended Kalman filtering algorithm for data fusion. This study utilized the constructed experimental platform to collect images and ultra-wideband ranging data in outdoor environments and experimentally validated the combined positioning method. The experimental results show that the algorithm outperforms individual UWB or loosely coupled combination positioning methods in terms of positioning accuracy. It effectively eliminates non-line-of-sight errors in UWB, improving the accuracy and stability of the combined positioning system.

## 1. Introduction

Location-based services (LBS) have gradually penetrated into all aspects of human life as a way of life. Accurate location information is a prerequisite for conducting LBS. At present, humans can rely on the Global Navigation Satellite System (GNSS) [1,2] to achieve sub-meter-level outdoor positioning accuracy; however, due to the inability of satellite signals to penetrate buildings and the high attenuation of GNSS signals, indoor positioning cannot be achieved using GNSS. How to achieve continuous indoor and outdoor positioning has become a research hotspot in the field of navigation and positioning, and it is also the trend of future positioning development. Both industry and academia are exploring high-precision and highly reliable positioning technologies in order to obtain accurate location information in indoor environments with complex electromagnetic and geographical environments. In 2011, Google relied on wireless fidelity (WiFi) and mobile communication base stations to release indoor maps, covering buildings such as shopping malls, supermarkets, airports, and stations; Apple relies on a large number of iBeacon devices and users’ iPhones to draw indoor maps with higher accuracy than Google; the University of Calgary in Canada combines sensors, public wireless signals (WiFi, Bluetooth), and indoor environmental features (such as magnetic field environments) to provide users with real-time and reliable indoor location. In addition, Samsung from South Korea, Spirit from Russia, Zetesis from Italy, University of Stuttgart in Germany, University of Seoul in South Korea, University of Antwerp in Belgium, and others have conducted research in the field of indoor positioning. In recent years, multi-sensor fusion positioning technology has achieved seamless continuous indoor and outdoor positioning, and has also become the mainstream positioning method in the field of navigation positioning.

Currently, there are several mainstream indoor positioning technologies, including WIFI, Bluetooth, ultra-wideband communication, inertial navigation system (INS), LiDAR, and vision [3,4]. Each of these technologies has its own advantages and disadvantages. INS positioning technology offers high short-term incremental data accuracy and strong autonomy. However, it suffers from severe error accumulation, drift, and high cost [5]. LiDAR can accurately measure the angle and distance of obstacles to generate an easily navigable environment map. However, it is highly dependent on the environment and the size of the scanned area, leading to poor stability [6]. Visual SLAM, which relies on cameras as positioning sensors, is a passive sensor that does not require pre-arrangement of the scene. It provides rich information in lightweight and inexpensive form and can be easily combined with other sensors. However, visual SLAM may encounter error accumulation issues during long-term or remote operation. Therefore, relying solely on one positioning method in complex environmental conditions makes it difficult to achieve high robustness and accuracy. Ultra-wideband (UWB) positioning is an active method with high bandwidth and strong penetration ability. It can provide absolute positioning services in environments where GNSS is unable to determine positioning [7]. UWB positioning technology adopts the principle of trilateral measurement, utilizing multiple base stations to construct a map for real-time positioning. It operates similarly to satellite positioning by observing the position of the base station for positioning. This ensures real-time performance without causing drift, enabling stable long-term operation [8]. By combining UWB positioning technology with visual SLAM, the map composed of base stations does not require correction through looping. The integrated navigation system can work in a global coordinate system, effectively complementing the visual system and ensuring stability [9,10].

Reference [11] proposes a tightly coupled fusion scheme for visual inertial density measurement (VIO) and UWB, where the measurement data from VIO and UWB are fused to obtain the robot’s posture. Reference [12] uses the position information output with the monocular vision ORB-SLAM algorithm and the positioning information obtained using UWB as measurement information, employing the extended Kalman filter (EKF) to fuse indoor positioning data. However, this article does not address the identification and elimination of non-line-of-sight (NLOS) errors in UWB. Reference [13] proposes a loosely coupled scheme for binocular VIO and UWB, utilizing the position information output with binocular VIO and UWB separately. The optimal position estimation of the robot can be obtained through data fusion using EKF. This scheme adopts a loosely coupled fusion method, and the robustness of the combined system is relatively low. Reference [14] proposed a method for EKF localization using inertial sensors, monocular vision, and UWB. This method utilizes UWB assisted visual inertial range synchronization and mapping to obtain improved drift free global six-degree-of-freedom attitude estimation. Although it achieves high accuracy, using three sensors increases cost and complexity of fusion.

Data fusion processing is the most important step in visual SLAM/UWB combination navigation solutions [15,16]. The methods of sensor information fusion mainly include the weighted average method, Kalman filtering method, Bayesian inference method, neural network algorithm, etc. The weighted average method is simple and intuitive, but it is difficult to obtain the optimal weighted average, and calculating the optimal weighted average value requires a lot of time. Describing Bayesian inference information as a probability distribution requires prior probability and likelihood function, and the analysis and calculation are complex. Neural network algorithms train and adjust network weights based on input data samples, but require a large amount of data during the training phase, resulting in poor real-time performance [17]. The extended Kalman filter (EKF) and unscented Kalman filter (UKF) are commonly used filtering strategies for nonlinear systems [18,19]. UKF uses a set of deterministic selected sigma points to approximate the probability distribution of system states, and further propagates them through nonlinear system models, resulting in higher order approximation accuracy than EKF. However, due to the presence of negative weights, UKF is considered unstable, especially for high-dimensional (over three) nonlinear systems [20]. EKF is one of the earliest nonlinear filters proposed, and D’Alfonso et al. found in their research that the performance of these two filters is comparable [21]. This may be due to insufficient nonlinearity of the model or the fact that the parameters have not been optimized yet [22]. The accuracy of location information directly affects the performance of indoor mobile robots. Chen et al. combined EKF with the least squares support vector machine to achieve accurate positioning [23]. In this method, EKF estimates the position and velocity information of the robot and uses the results for training the compensation model. Although EKF is not suitable for non-Gaussian processes, it has significant advantages in computing speed and resource consumption, and is easy to implement and run online on computers. Therefore, in this article, the extended Kalman filtering algorithm was chosen to handle nonlinear problems for sensor information fusion [24].

In a visual/UWB fusion positioning system, obstacles can block UWB pulse signals, leading to phenomena such as signal reflection, refraction, and penetration, resulting in non-line-of-sight (NLOS) environments and NLOS errors. NLOS errors are the main source of error in UWB positioning solutions, significantly reducing the accuracy and stability of UWB positioning systems. There are two main approaches to handling NLOS errors: robust filtering methods and data signal feature identification methods. In terms of robust filtering methods, reference [25] proposes an algorithm that uses colored noise adaptive Kalman filtering. While this algorithm produces significant results, it is complex and computationally demanding. Reference [26] investigates the CC-KF algorithm, which divides the TOF signal propagation path into three types and adjusts the parameters of the Kalman filter based on the different paths. This method can effectively eliminate different types of NLOS errors but may introduce larger errors due to path errors. In terms of data signal feature identification methods, reference [27] proposes an NLOS identification method based on signal strength. This method analyzes and identifies NLOS based on the characteristics of received signals. It has significant identification performance and wide applicability but requires prior observation and feature extraction of NLOS and line-of-sight (LOS) samples, resulting in a significant amount of work. Reference [28] directly extracts feature parameters from signals and uses least squares support vector machines for NLOS identification. However, this method has high complexity and is time-consuming. While robust filtering methods mostly provide weak constraints on NLOS errors and are difficult to completely eliminate them, data signal feature identification methods are more direct and accurate but have higher algorithm complexity. Therefore, it is necessary to design more accurate and practical methods for NLOS error identification and suppression. One approach is to monitor faulty data based on the consistency of redundant information at the receiver end and directly remove faulty data from the raw data.

Based on the above analysis, this article proposes a visual/UWB fusion localization method. It uses the displacement increment of visual measurement method (VO) and the ranging information of UWB as measurement values, and uses the EKF algorithm for parameter estimation. Considering that measurement noise is easily affected by complex environments, threshold detection and adaptive measurement noise estimator are introduced to suppress the impact of outliers and time-varying measurement noise on filter performance. This fusion positioning method utilizes visually obtained distance information to eliminate UWB NLOS errors, effectively suppressing the impact of NLOS errors on the fusion system and improving the positioning accuracy of the fusion system.

## 2. Principles of the Visual and UWB Localization Algorithms

### 2.1. Principles of the Visual SLAM Localization Algorithm

The ORB-SLAM2 algorithm is used for visual positioning, and the front-end visual odometer is based on the “Oriented FAST” key points and the BRIEF descriptor, with the aim of achieving feature point extraction and matching [29]. The back end is based on a nonlinear optimized BA visual SLAM system. It divides the traditional visual SLAM algorithm into three threads: position tracking, local mapping, and loop closing. The flowchart of the algorithm is shown in Figure 1. In the RGB-D mode, the tracking thread is responsible for real-time pose localization, tracking, and optimization processing based on the provided feature point depth information. The local mapping thread creates new map points through the obtained keyframes and removes points outside the map. The pose of the keyframes is locally optimized via BA, and redundant keyframes and map points are deleted. The closed-loop detection thread uses a mathematical model to evaluate the similarity of adjacent keyframes and determine the closed-loop situation of keyframes, which helps to reduce the cumulative drift of trajectories. This article selects an RGB-D camera as the image input source. Compared to monocular and binocular cameras, the RGB-D camera can simultaneously capture color images and corresponding depth maps. Not only can it solve the problem of scale uncertainty in monocular vision, but it can also eliminate the tedious steps of calculating the parallax between left and right cameras in binocular vision, reduce computer computation, and ensure real-time requirements.

### 2.2. Principle of the Ultra-Wideband (UWB) Positioning Algorithm

Ultra-wideband communication is a wireless carrier communication technology that uses nanosecond non-sinusoidal narrow pulses to transmit data, occupying a wide spectral range. Due to its simple system structure, high transmission rate, and low functionality, it is widely used in positioning technology.

The commonly used positioning methods for UWB technology include TOA (time of arrival), TDOA (time difference of arrival), TWR (time of flight ranging), etc. The principle of TOA positioning is to calculate the distance of signal propagation by measuring the propagation time of the signal from the transmitting source to the receiver. It uses multiple receivers (anchors or base stations) to simultaneously measure the arrival time of the signal and calculate the position of the target via multilateral positioning algorithms (such as triangulation). The TDOA positioning principle is based on using the difference in the signal arrival time to calculate the target position. In TDOA positioning, at least three receivers are required for measurement. By measuring the time difference between signals reaching different receivers, the position of the target relative to these receivers can be calculated. The principle of TWR positioning is to calculate distance based on the flight time of the signal. TWR calculates the distance of signal propagation by measuring the flight time from the transmitter to the receiver, combined with the propagation speed. Multiple TWR measurements can be used for multilateral positioning and calculating target positions.

In our experiments, TOA positioning was selected. Unlike other positioning methods, TOA does not require advanced hardware or complex signal processing algorithms, and it is relatively easy to implement. It has high accuracy and a relatively simple implementation method. Moreover, TOA positioning is based on time measurements, and its measurement accuracy can reach the sub-nanosecond level or better. Therefore, it has high positioning accuracy and a good ability to suppress multipath effects. The schematic diagram is as follows:

Assuming the target generates a signal at time $t_{0}$ and, at time $t_{R}$, reaches the receiver, the propagation time of the signal from the transmitting source to the receiver Δt (TOA) can be expressed as:(1)$$\Delta t=t_{R}-t_{0}$$

The distance d of signal propagation is as follows:(2)d=c×Δt
where c represents the propagation speed of the signal (usually approximately the speed of light).

In this way, by measuring the time of arrival (TOA) and signal propagation speed of the signal, the distance of signal propagation can be calculated, thereby achieving target positioning. In practical applications, multiple receivers (anchors or base stations) are usually used to calculate the position of the target through multilateral positioning algorithms (such as triangular positioning algorithms), as shown in Figure 2.

When installing and deploying base stations 1, 2, and 3, their positions are fixed and known. For base stations 1 ($x_{1}$, $y_{1}$), 2 ($x_{2}$, $y_{2}$), and 3 ($x_{3}$, $y_{3}$), the coordinates of the required positioning labels are ${\;R}_{0\;}(x_{0}$, $y_{0}$). The location of an unknown tag $R_{0\;}(x_{0}$, $y_{0},z_{0}$) can be computed via trilateration, i.e., Pythagoras theorem:(3)$$\left\{\begin{array}{c} x_{0}^{2}+y_{0}^{2}+z_{0}^{2}=d_{1}^{2}\\ {\left(x_{0}-x_{2}\right)}^{2}+y_{0}^{2}+z_{0}^{2}=d_{2}^{2}\\ {\left(x_{0}-x_{3}\right)}^{2}+{\left(y_{0}-y_{3}\right)}^{2}+z_{0}^{2}=d_{3}^{2} \end{array}\right.$$
where $d_{i}$ is the measured distance (radius of a circle or sphere) between the ith anchor and the tag, and $(x_{0}$, $y_{0}$) is the interested unknown location of the tag.

Through the deformation of formulas, we obtain the following:(4)$$x_{0}=\frac{d_{1}^{2}-d_{2}^{2}+x_{2}^{2}}{2x_{2}}$$
(5)$$\;y_{0}=\frac{d_{1}^{2}-d_{3}^{2}+x_{3}^{2}+y_{3}^{2}-{2x}_{0}x_{3}}{2y_{3}}$$
(6)$$\;z_{0}=\pm \sqrt{d_{1}^{2}-x_{0}^{2}+y_{0}^{2}}$$

To determine the ambiguous solutions from (6), the information from the fourth anchor is necessary in 3D trilateration. There are three constraints in the geometry of trilateration: (i) the first anchor (A1) should be located in the origin of a coordinate system, i.e., (0, 0, 0) in 3D Cartesian coordinate, (ii) the second anchor should be located on the X-axis, and (iii) the height of the anchors (Z-value) should be the same for all anchors. In an arbitrary system set-up, the first constraint can be accomplished by subtracting the value of the first anchor (A1) from all the three available known anchors including itself. The second constraint can be accomplished by projecting the second anchor’s value (A2) onto the X-axis.

The generic spherical equation for true-range multilateration can be represented in 3D as follows:(7)$$\;d_{i}^{2}={\left(x_{i}-x_{0}\right)}^{2}+{\left(y_{i}-y_{0}\right)}^{2}+{\left(z_{i}-z_{0}\right)}^{2}$$
where, $d_{i}$ is the distance (range or radius of a sphere) between the coordinates of the ith anchor and the tag.

## 3. Fusion Localization Algorithm

### 3.1. UWB Non-Line-of-Sight Error Identification

When NLOS error occurs in UWB positioning, the ranging value of UWB will undergo significant changes. If an NLOS error is not processed, the accuracy and stability of the visual/UWB combined positioning system will be greatly reduced. Visual SLAM obtains the pose of each keyframe of the camera by preprocessing, initializing, estimating pose, and tracking local maps on image sequences. Under normal tracking conditions, the displacement increment of visual SLAM is relatively accurate. Therefore, this article uses the displacement increment of visual SLAM to identify NLOS errors in UWB ranging values. Based on the coordinate increment of visual SLAM and the coordinates of the previous epoch in the combination system, calculate the coordinates of the current epoch combination system, and then calculate the distance between the combination system and the UWB reference station:(8)$$\left\{\begin{array}{c} x_{k}^{c}=x_{k-1}+∆x_{k}^{c}\\ y_{k}^{c}=y_{k-1}+∆y_{k}^{c} \end{array}\right.$$
(9)$$d_{k,i}^{c}=\sqrt{{\left(x_{k}^{c}-x_{i}^{b}\right)}^{2}+{\left(y_{k}^{c}-y_{i}^{b}\right)}^{2}},\;i=1,2,3\cdots ,N$$

In the formula, $\left(x_{k}^{c},y_{k}^{c}\right)$ is the coordinate of the combined system visually solved at time k; $\left(x_{k-1},y_{k-1}\right)$ is the coordinate of the combined system at time k−1; ($∆x_{k}^{c},∆y_{k}^{c})$ is the visual coordinate increment at time k; and $d_{k,i}^{c}$ is the distance between the combination system visually calculated at time k and the i-th UWB reference station. ${(x}_{i}^{b},y_{i}^{b})$ is the known coordinate of the i-th UWB reference station; N is the number of UWB reference stations, N ≥ 3. Subtract the ranging value of UWB from the visual calculated distance value and compare the difference with the set threshold, as follows:(10)$$d_{k,i}^{u}=\sqrt{{\left(x_{k}^{u}-x_{i}^{b}\right)}^{2}+{\left(y_{k}^{u}-y_{i}^{b}\right)}^{2}}+v_{k,i},\;i=1,2,3\cdots ,N$$
(11)$$∆d_{k,i}=d_{k,i}^{c}-d_{k,i}^{u}$$
(12)$$\left\{\begin{array}{l} \left|∆d_{k,i}\right|\geq L,NLOS\\ \left|∆d_{k,i}\right|<L,LOS \end{array}\right.$$

In the formula, $d_{k,i}^{u}$ is the observation distance between the UWB mobile station and the i-th UWB reference station in the combination system at time k. ${(x}_{k}^{u},y_{k}^{u})$ is the coordinate of the UWB mobile station at time k; $v_{k,i}$ is the measurement noise sequence of UWB, $∆d_{k,i}$ is the distance difference; and L is the threshold, L > 0.

When NLOS error occurs, UWB ranging values will be affected by the NLOS error and cause jumps, resulting in range differences fluctuating within a certain range. Therefore, this article first provides an initial empirical threshold and then amplifies the average of n distance differences by an appropriate multiple as the dynamic testing threshold for distance differences. When $\left|∆d_{k,i}\right|$ is greater than or equal to threshold, the distance measurement value from the combined system to the i-th UWB reference station at the current time is removed. If $\left|∆d_{k,i}\right|$ is less than the threshold, the current ranging value is retained.

According to the visual SLAM and UWB ranging accuracy levels, an initial threshold of 0.3 m is given. If $\left|∆d_{k,i}\right|$ ≥ 0.3 m, then exclude $∆d_{k,i}$ and ${\;d}_{k,i}^{u}$. If $\left|∆d_{k,i}\right|$ < 0.3 m, then $d_{k,i}^{u}$ join the combined positioning process while retaining $∆d_{k,i}$ until n $∆d_{k,i}s$ are retained, stop using the initial threshold, and take n as 20. Take the average of these 20 distance differences and multiply the obtained average by three times as the dynamic test threshold. Afterwards, a dynamic testing threshold is used to distinguish whether there is an NLOS error in $d_{k,i}^{u}$, and the dynamic testing threshold is continuously updated based on $∆d_{k,i}$.

### 3.2. Algorithm for Combined Positioning

Compared with loose coupling, the tight coupling fusion localization of UWB and vision has strong environmental adaptability and better robustness. Considering that the tight coupling mode has lower delay and faster response speed, it can better adapt to the NLOS detection method proposed in the previous section by obtaining and processing real-time information from the receiver and transmitter. Therefore, this paper chooses the tight coupling mode.

According to Newton’s second law of motion, the state dynamic model of navigation and tracking systems is usually assumed to be linear. Therefore, in the visual/UWB combined positioning system of this article, nonlinearity only appears in the measurement function in (10). This means that the state model in EKF remains exactly the same as the standard KF.

The state equation of the combination algorithm is as follows:(13)$$X_{k}=FX_{k-1}+w_{k-1}$$
where $X_{k}={\left[x_{k},y_{k},v_{k,\;x},v_{k,y}\right]}^{T}$ is the state vector, $x_{k}$,$y_{k}$ represents the coordinates in the x and y directions of the combined system at time k, and $v_{k,x},v_{k,y}$ represents the velocities in the x and y directions of the combined system at time k; $F=\left[\begin{array}{cccc} 1 & 0 & ∆t & 0\\ 0 & 1 & 0 & ∆t\\ 0 & 0 & 1 & 0\\ 0 & 0 & 0 & 1 \end{array}\right]$ is the state transition matrix, where ∆t is the sampling interval; $w_{k-1}$ is a sequence of process noise.

When using the EKF algorithm, the time update process is as follows:(14)$${\hat{X}}_{k}^{-}=F{\hat{X}}_{k-1}$$
(15)$$P_{k}^{-}=FP_{k-1}F^{T}+Q_{k}$$

In the formula, ${\hat{X}}_{k}^{-}$ is the prior predicted value of the state at time k; $P_{k}^{-}$ is the prior estimation matrix of error covariance; and $Q_{k}$ is the covariance matrix of process noise at time k.

According to Formula (10), the UWB positioning observation equation is
(16)$$d_{k,i}^{u}=\sqrt{{\left(x_{k}^{u}-x_{i}^{b}\right)}^{2}+{\left(y_{k}^{u}-y_{i}^{b}\right)}^{2}}+v_{k,i},\;i=1,2,3\cdots ,N$$

The true horizontal distance between UWB mobile station and reference station is
(17)$$d_{k,i}=\sqrt{{\left(x_{k}^{u}-x_{i}^{b}\right)}^{2}+{\left(y_{k}^{u}-y_{i}^{b}\right)}^{2}}\;i=1,2,3\cdots ,N$$

The approximate location $\left(x_{k0},y_{k0}\right)$ of the UWB mobile station in the *k*-th epoch is calculated using the previous epoch’s location information. The first-order Taylor expansion of Equation (17) at $\left(x_{k0},y_{k0}\right)$ yields
$$d_{k,i}=d_{k0,i}+\frac{\partial d_{k,i}}{\partial x_{k}}dx_{k}+\frac{\partial d_{k,i}}{\partial y_{k}}dy_{k}$$

In the formula,
$$d_{k0,i}=\sqrt{{\left(x_{k0}-x_{i}^{b}\right)}^{2}+{\left(y_{k0}-y_{i}^{b}\right)}^{2}}\;i=1,2,3\cdots ,N$$

So, the linearized equation is
(18)$${\tilde{d}}_{k,i}^{u}=d_{k0,i}+\varepsilon_{k,i}^{u}=d_{k0,i}+\frac{\partial d_{k,i}}{\partial x_{k}}dx_{k}+\frac{\partial d_{k,i}}{\partial y_{k}}dy_{k}+\varepsilon_{k,i}^{u}$$

Measurement equation is as follows:(19)$${\tilde{d}}_{k,i}^{u}-d_{k0,i}^{u}=\frac{x_{k0}-x_{i}^{b}}{\sqrt{{\left(x_{k0}^{u}-x_{i}^{b}\right)}^{2}+{\left(y_{k0}^{u}-y_{i}^{b}\right)}^{2}}}dx_{k}+\frac{y_{k0}-y_{i}^{b}}{\sqrt{{\left(x_{k0}^{u}-x_{i}^{b}\right)}^{2}+{\left(y_{k0}^{u}-y_{i}^{b}\right)}^{2}}}dy_{k}+\varepsilon_{k,i}^{u}$$

In the formula, ${\tilde{d}}_{k,i}^{u}$ is the distance measurement value identified via NLOS; $\left(x_{k0},y_{k0}\right)$ is the approximate coordinates of the combined system at time k; $d_{k0,i}^{u}$ is the approximate distance between the combined system at time k and the i-th UWB reference station; and $\varepsilon_{k,i}^{u}$ is linearized noise.

Equation (8) can be expressed as
(20)$$\left\{\begin{array}{c} x_{k-1}^{c}+∆x_{k}^{c}-x_{k0}=dx_{k}+\varepsilon_{x_{k}}^{c}\\ y_{k-1}^{c}+∆y_{k}^{c}-y_{k0}=dy_{k}+\varepsilon_{y_{k}}^{c} \end{array}\right.$$

In the formula, $\varepsilon_{x_{k}}^{c}$,$\varepsilon_{y_{k}}^{c}$ represents the linearized noise in the x and y directions at time k.

The measurement equation for a composite system is
(21)$$Z_{k}=HX_{k}+v_{k}$$

In the formula, $Z_{k}=\left[\begin{array}{ccc} {\tilde{d}}_{k,i}^{u}-d_{k0,i}^{u} & x_{k-1}^{c}+∆x_{k}^{c}-x_{k0} & y_{k-1}^{c}+∆y_{k}^{c}-y_{k0} \end{array}\right]$ is a measurement vector.
$$H=\left[\begin{array}{cccc} \frac{x_{k0}-x_{i}^{b}}{\sqrt{{\left(x_{k0}^{u}-x_{i}^{b}\right)}^{2}+{\left(y_{k0}^{u}-y_{i}^{b}\right)}^{2}}} & \frac{y_{k0}-y_{i}^{b}}{\sqrt{{\left(x_{k0}^{u}-x_{i}^{b}\right)}^{2}+{\left(y_{k0}^{u}-y_{i}^{b}\right)}^{2}}} & 0 & 0\\ 1 & 0 & 0 & 0\\ 0 & 1 & 0 & 0 \end{array}\right]$$
is a measurement matrix, and $v_{k}={\left[\begin{array}{ccc} \varepsilon_{k,i}^{u} & \varepsilon_{x_{k}}^{c} & \varepsilon_{y_{k}}^{c} \end{array}\right]}^{T}$ is a linearized measurement noise sequence.

The measurement update process of the EKF algorithm is as follows:(22)$$K_{k}={P_{k}^{-}H_{k}^{T}\left(H_{k}P_{k}^{-}H_{k}^{T}+R_{k}\right)}^{-1}$$
(23)$${\hat{X}}_{k}={\hat{X}}_{k}^{-}{+K}_{k}\left(Y_{k}-H{\hat{X}}_{k}^{-}\right)$$
(24)$$P_{k}=\left(I-K_{k}H_{k}\right)P_{k}^{-}$$

In the formula, $K_{k}$ is the filtering gain matrix at time k; $R_{k}$ is the covariance matrix of the observed noise at time k; and I is the identity matrix.

Given the initial value of ${\hat{X}}_{0}=E\left[X_{0}\right]P_{0}=E[\left(X_{0}-{\hat{X}}_{0}\right)-{\left(X_{0}-{\hat{X}}_{0}\right)}^{T}]$, $R_{0}=E[v_{0}v_{0}^{T}]$. Based on the state equation and measurement equation, the EKF algorithm is used to update the state and measurement and to obtain the positioning information of the composite system.

### 3.3. Measurement Noise Estimation and Threshold Judgment

For combined positioning systems, the system is susceptible to abnormal interference during maneuvering, and measurement inevitably involves errors such as typical outliers in observations and non-Gaussian characteristics of noise statistics. Therefore, a method is needed to effectively handle and eliminate the aforementioned errors involved. Reference [30] proposed a new adaptive robust strategy based on Mahalanobis distance to weaken the influence of outlier model bias and outliers in measurements; reference [31] uses the Variance Shift Outlier Model (VSOM) to detect faults in raw pseudo range data. According to the magnitude of the relevant variance shift, measurements are partially excluded or included in the estimation process. This method can accurately detect and identify faults that occur during the navigation process. In the estimation process, weighting faults using appropriate weighting factors can ensure performance and accuracy. Reference [32] proposes an innovative saturation mechanism that applies the saturation function to the innovation process of correcting state estimation in EKF. Therefore, when outliers occur, the innovation of distortion is saturated to avoid disrupting state estimation. The characteristic of this mechanism is used to adaptively adjust the saturation boundary, making EKF robust to outliers. Reference [33] established a stochastic model of an integral Kalman filter using analysis of variance and non-holonomic constraints. In a loosely and tightly coupled integration mode, KF with fault detection and troubleshooting capabilities is adopted to reduce the adverse effects of abnormal GNSS data. Reference [34] presents an adaptive UKF with noise statistic estimator to overcome the limitation of the standard UKF. According to the covariance matching technique, the innovation and residual sequences are used to determine the covariance matrices of the process and measurement noises. The proposed algorithm can estimate and adjust the system noise statistics online, and thus enhance the adaptive capability of the standard UKF.

Therefore, this article adds a Sage Husa noise estimator and threshold judgment on the basis of EKF. On the one hand, the noise estimator can continuously adjust the measurement noise intensity. On the other hand, by adding a threshold judgment mechanism and deleting measurement data that are severely offset from the actual position, optimization of the filter can be achieved.

Due to the traditional EKF requiring measurement noise to follow a Gaussian distribution of zero mean on the noise assumption $Q_{k}\sim N(0,Q)$, $R_{k}\sim N\left(0,R\right)$, where $Q_{k}$ should be related to the system prediction model, $R_{k}$ is mainly related to sensor measurement data. In practical scenarios, such as SLAM and UWB positioning, the measurement noise may not fully conform to the Gaussian distribution and it may be affected by the environment, resulting in increased or even divergent estimation errors. Therefore, this article introduces the Sage Husa noise estimator and threshold judgment mechanism based on traditional EKF to optimize the performance of the filter.

The Sage Husa noise estimator is an adaptive noise estimation technique that continuously adjusts the intensity of measurement noise. In each iteration, by analyzing the measurement data, the actual measurement noise intensity is estimated and applied to the noise model in EKF to reflect the actual measurement error more accurately. The threshold judgment mechanism analyzes the estimated values of the filter output to determine whether there is significant deviation from the actual position of the measurement data, preventing these abnormal data from interfering with the filter estimation, and thus achieving the optimization of the filter.

When $w_{k}$ is fixed, the Sage Husa algorithm is used to estimate the covariance of system measurement noise. For the k measurement, first, the measurement residual is calculated as follows:(25)$$e_{k}=y_{k}-H{\hat{X}}_{k}^{-}$$
where $y_{k}$ is the actual measurement value, H is the state observation matrix, and ${\hat{X}}_{k}^{-}$ is the prior state prediction value. Then, the covariance ${\hat{R}}_{k}$ [35] of the measurement noise is estimated using the following formula:(26)$${\hat{R}}_{k}=\left(1-d_{k}\right){\hat{R}}_{k}+d_{k}\left(e_{k}e_{k}^{T}-H{\hat{X}}_{k}^{-}H^{T}\right)$$
where ${\hat{R}}_{k}$ is the k-th estimated measurement noise covariance matrix, $d_{k}$ is the adaptive weight, and $d_{k}=\frac{1-b}{\;1+b^{k+1}}$. b is the forgetting factor, ranging from 0.95 to 0.99, representing the estimated weight before forgetting it. The specific value can be obtained through experiments.

To eliminate abnormal measurement data, before fusion, the measured values are compared with the estimated values of the state vector, and the difference between the two is subtracted from the pre-set threshold. If the threshold is exceeded, the previous state estimation is used instead.

After combining the threshold judgment, Equation (22) in EKF is modified to
(27)$$K_{k}=\left\{\begin{array}{c} 0,\;\left|e_{k}\right|>thresh\\ {{\;P}_{k}^{-}H_{k}^{T}\left(H_{k}P_{k}^{-}H_{k}^{T}+{\hat{R}}_{k}\right)}^{-1},\left|e_{k}\right|<thresh \end{array}\right.$$
where the threshold is the pre-selected threshold. The main steps of the final fusion localization algorithm are shown in Figure 3.

## 4. Testing and Analysis

### 4.1. Construction of the Visual/UWB Platform

The camera used on the mobile positioning platform is Intel’s Realsense-D455 model camera (Intel, Santa Clara, CA, USA), with a resolution of 1280 pixels by 720 pixels. The sampling frequency was set to 10 Hz. The UWB used was the Decawave’s DW1000 communication and ranging module (DecaWave, Dublin, Ireland), with a bandwidth of 3.5–6.5 GHz and a data sampling frequency of 10 Hz. In order to accurately obtain the system error of each anchor point of the UWB, a mobile calibration platform was constructed in this experiment, as shown in Figure 4. The platform consists of three main parts: aluminum profile chassis, a control box, and a synchronous belt. Both the UWB and the camera are fixed on the moving slider of the synchronous belt of the mobile calibration platform. By controlling the synchronous belt via the control box, the speed, acceleration, and spatial coordinate values of the two sensors can be accurately obtained. Time tags are added to the data collected via UWB positioning and the camera, ensuring that these time tags are consistent with the system time of the computer. Data are collected from both sensors on the laptop at the same time.

### 4.2. Correction of UWB Positioning System

The TOA (time of arrival) algorithm used in this article usually includes two stages: a ranging stage and a localization stage. Due to environmental and other external factors, there may be some errors in the distance measured via UWB positioning. Therefore, in order to accurately obtain the measurement error of UWB and improve subsequent positioning accuracy, this section of the experiment collected distance information of four UWB anchor points working simultaneously.

First, four UWB anchor points are fixed within the flat area of the large scene, as shown in Figure 5. The label and anchor are kept at the same height. The target labels are placed at different positions and distances within the range of 0–100 m for the distance measurement, with 20 groups every 5 m. Data are collected for 2 min for each group, including 400 measurement data. The measurement data of a laser rangefinder with a measurement accuracy of 0.001 m as a reference value are compared with the collected measurement data.

Figure 6 shows the relationship between the absolute error and distance of each anchor point, both before and after correction. As shown in Figure 6, there is a difference in the absolute error between the measured values of each anchor point at the same true distance and the environment, which is caused by hardware factors. Moreover, the measurement error of each anchor point fluctuates between 0.010 and 0.040 m at different distances. This indicates that there are certain systematic and random errors in UWB distance measurement.

The system error remains relatively stable throughout the entire measurement system process. In order to eliminate the impact of the system error on the positioning results, the system is calibrated by collecting reference point data at a known distance, establishing a system error compensation model, and applying it to distance measurement. The corrected absolute error is shown with the blue line in Figure 6. A comparison of measurement error data before and after correction is shown in Table 1. The random error is processed using a Kalman filter, and the results before and after processing are shown in Figure 7.

The analysis shows that the average measurement error after correction has decreased by 5.9 cm and the standard deviation has decreased by 0.08 compared to the values before correction. This indicates that this pre-experiment effectively improved the performance of the UWB positioning system, making its distance measurement more accurate and stable.

By conducting the pre-experiments described in this section, we obtained distance error compensation models for UWB during the positioning process. These models can be used in subsequent positioning experiments to reflect the distance measurement noise of UWB. This is very helpful for the design and performance evaluation of subsequent localization algorithms.

### 4.3. Outdoor Positioning Test and Analysis

We conducted positioning experiments in both obstacle-free and obstacle-rich environments. The obstacle-free environment was a volleyball court outside the teaching building, while the obstacle-rich environment was a forest next to the laboratory building. In the obstacle-rich environment, tree branches and trunks served as obstacles during the positioning process, while the small field had no obstacles. The experiments were conducted around 14:00 when the lighting conditions were good, facilitating the experiment.

For the experiments, we used four UWB modules, a self-made mobile positioning platform, an Intel RealSense depth camera, and a Xiaomi laptop for fusion positioning. The four UWB modules were placed at the four corners of the field as anchor nodes. The coordinates of the four UWB reference stations were obtained in advance using a total station and a laser rangefinder. The experimental setup and layout are shown in Figure 8. The mobile positioning platform moved a distance of 3 m, with a conveyor belt speed of 0.105 m/s. The platform had a UWB mobile tag attached to it, and the camera moved along a fixed trajectory at a constant speed.

Figure 9 shows the comparison between UWB, visual/UWB loosely coupled models, and visual/UWB-EKF tightly coupled fusion models with ground truth trajectories. In a good line-of-sight (LOS) scene, visual/UWB-EKF achieves centimeter level positioning accuracy, and the estimated trajectory closely matches the reference trajectory.

Throughout the entire experimental process, compared with the other two methods, the visual/UWB-EKF tightly coupled fusion method consistently showed smaller positioning errors. Table 2 provides the error statistics of these three positioning methods in the x and y directions. In Table 2, RMSE represents root mean square error, and MAX represents maximum error value. See also Figure 10.

From Table 2, we can observe the following:

The maximum error of the visual/UWB-EKF fusion method in the x direction is 5.7 cm, and the maximum error in the y direction is 3.3 cm. The positioning accuracy can be continuously maintained at around 4 cm. Compared with the other two methods, this fusion method has better robustness, higher positioning accuracy, and stronger reliability.

From Figure 11, it can be observed that in the presence of obstacles, the visual/UWB-EKF fusion method effectively identifies and eliminates UWB NLOS errors by utilizing visual displacement increments, resulting in a trajectory closely following the reference trajectory. The inclusion of UWB ranging information also helps reduce visual odometry error accumulation. The maximum positioning error is 14.6 cm. Compared to the other two methods, it is evident that the visual/UWB-EKF fusion method produces good results, effectively mitigating the impact of UWB NLOS errors on positioning results and reducing trajectory drift. The visual/UWB-EKF fusion method’s positioning trajectory aligns better with the reference trajectory, demonstrating strong reliability.

It can be observed that, compared to the loosely coupled mode, the visual/UWB-EKF fusion system reduces the root mean square error in the x direction from 0.106 m to 0.076 m, and, in the y direction, it reduces it from 0.086 m to 0.082 m. The maximum positioning error in the x direction decreases from 0.195 m to 0.146 m, and, in the y direction, it decreases from 0.148 m to 0.122 m. The experimental results demonstrate that the visual/UWB-EKF fusion model effectively addresses the issue of large trajectory errors in UWB positioning and suppresses the impact of UWB NLOS errors on the fusion system’s positioning accuracy. See Table 3 and Figure 12.

The experimental results indicate that the proposed fusion positioning method is feasible. It effectively solves the problem of error accumulation in visual odometry, improves the stability of the VO algorithm, and addresses UWB NLOS errors, enhancing the robustness of UWB positioning results. The fusion positioning accuracy can reach sub-decimeter level.

## 5. Conclusions

This article introduces a fusion localization algorithm that combines visual simultaneous localization and mapping (SLAM) technology with ultra-wideband (UWB) technology to address the issues of low accuracy and poor stability in single localization methods for mobile robots. The algorithm is based on the traditional extended Kalman filter (EKF) and utilizes visual displacement increment to design a UWB non-line-of-sight error discrimination method. It also adds measurement noise estimator and threshold detection, improving the stability of the fusion positioning system in complex environments. The experimental results show that the visual/UWB-EKF combination method reduces error accumulation, eliminates the NLOS impact of UWB, and improves the overall positioning accuracy and robustness compared to a single-sensor and loosely coupled combination.

However, due to various limitations, there are still aspects in the designed positioning system that can be further explored. This article outlines the following aspects for reference:(a)This article overlooks the small spatial offset generated using the UWB label device on the mobile platform in the design. It uses the ranging value obtained from the UWB sensor directly as the distance between the camera and the UWB anchor point. This may introduce errors between the calculated position and the actual position, which can impact the algorithm’s accuracy. Addressing this issue in the future could provide greater flexibility for the mechanical setup of the system. Additionally, over time, UWB anchor degradation may occur, and this should be carefully monitored to ensure the accuracy of the obtained position.(b)The experiments in this article are conducted in a small field, leading to fewer accumulated errors. While the errors caused by the sensor itself are considered during data modeling, the impact of accumulated errors is closely related to the experimental site and the system’s operation time. Therefore, the results obtained in different environments may vary.

Future research will address these limitations to make this study more comprehensive.

## Figures and Tables

**Figure 1 sensors-24-01710-f001:**
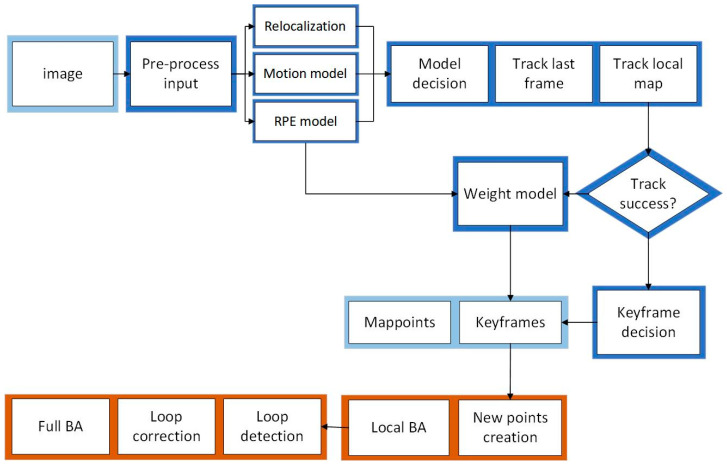
ORB-SLAM2 algorithm flowchart.

**Figure 2 sensors-24-01710-f002:**
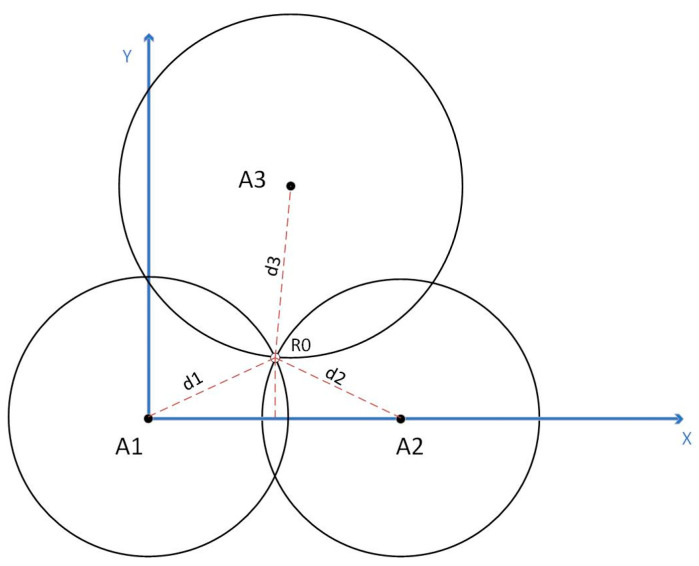
Illustration of trilateration algorithms in 2D.

**Figure 3 sensors-24-01710-f003:**
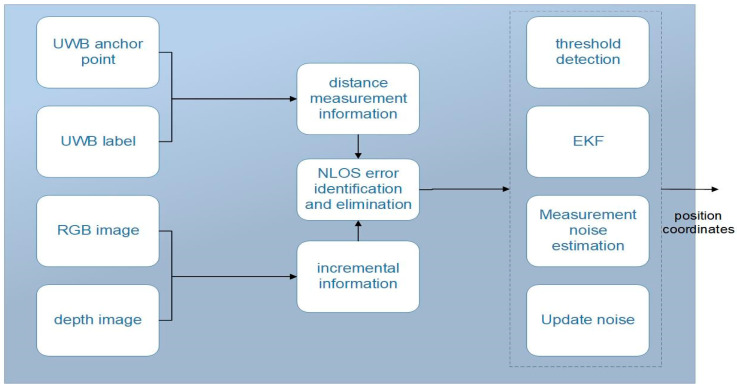
Framework diagram of the fusion localization algorithm.

**Figure 4 sensors-24-01710-f004:**
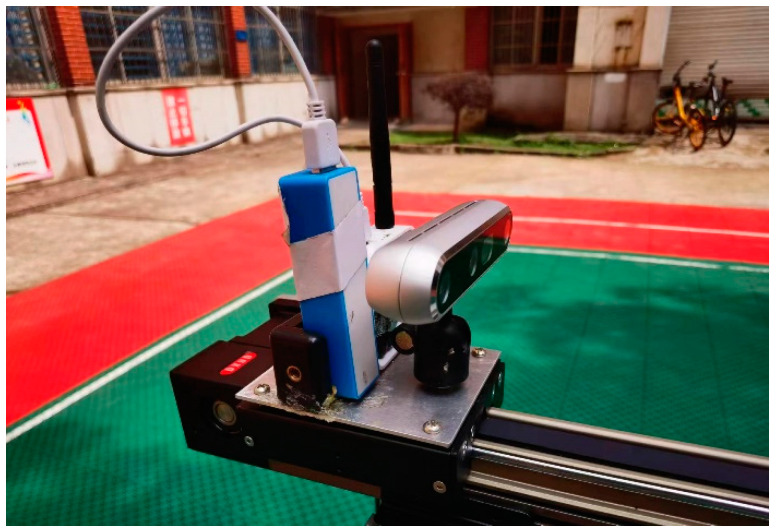
Mobile calibration platform.

**Figure 5 sensors-24-01710-f005:**
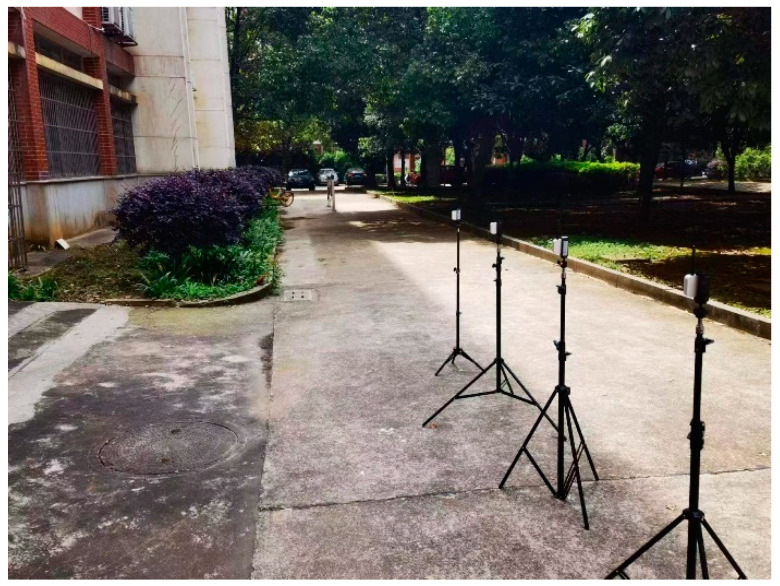
Correction test.

**Figure 6 sensors-24-01710-f006:**
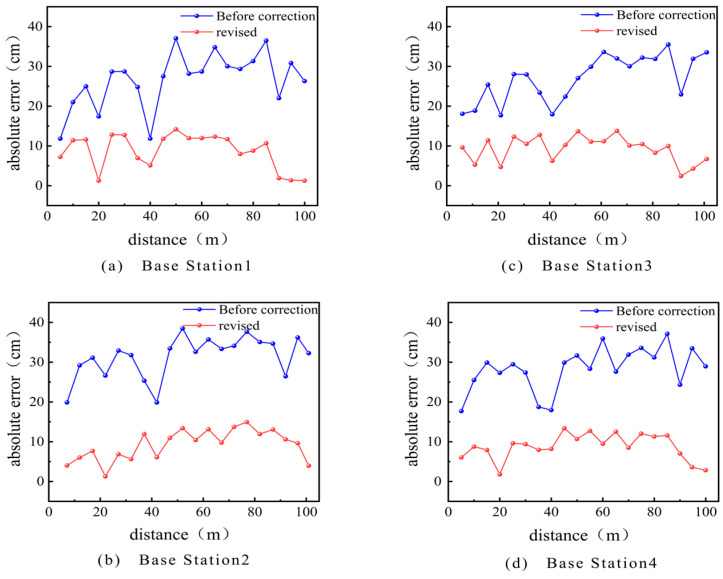
Absolute error of UWB anchor ranging before and after correction.

**Figure 7 sensors-24-01710-f007:**
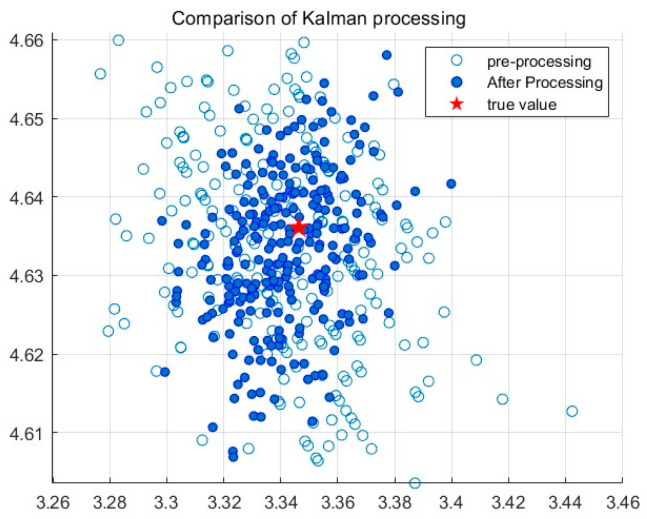
Comparison before and after Kalman filtering.

**Figure 8 sensors-24-01710-f008:**
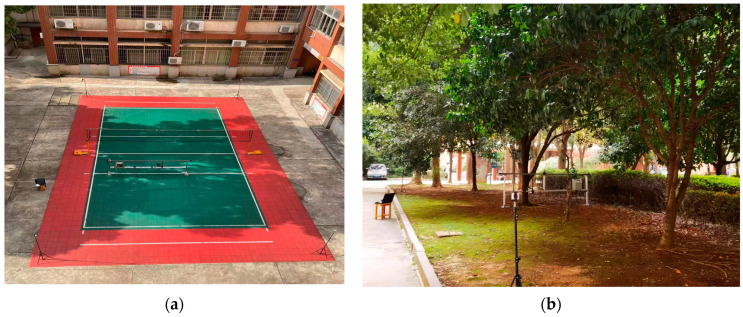
(**a**) Line-of-sight scenes; (**b**) non-line-of-sight scenes.

**Figure 9 sensors-24-01710-f009:**
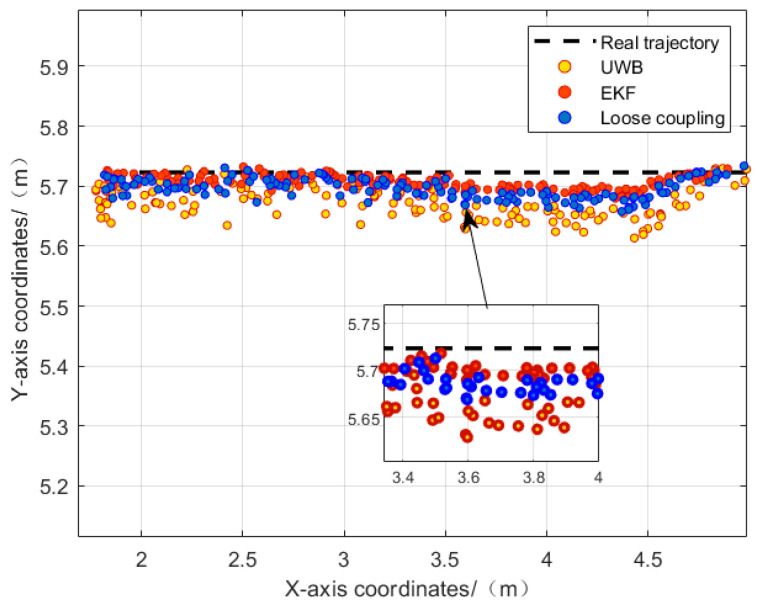
Comparison between solution trajectory and reference trajectory in the line-of-sight state.

**Figure 10 sensors-24-01710-f010:**
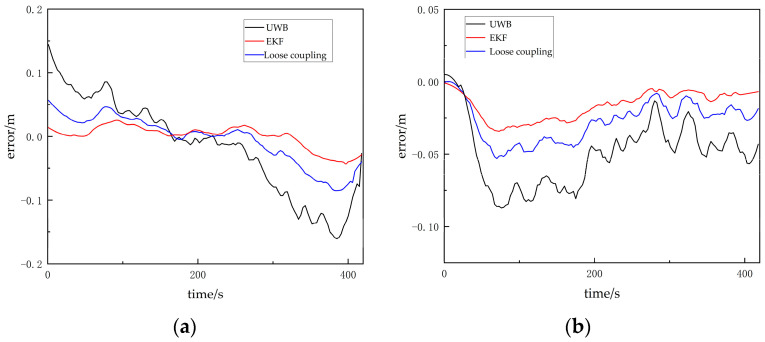
(**a**) Error in the x direction in line-of-sight scenarios; (**b**) error in the y direction in line-of-sight scenarios.

**Figure 11 sensors-24-01710-f011:**
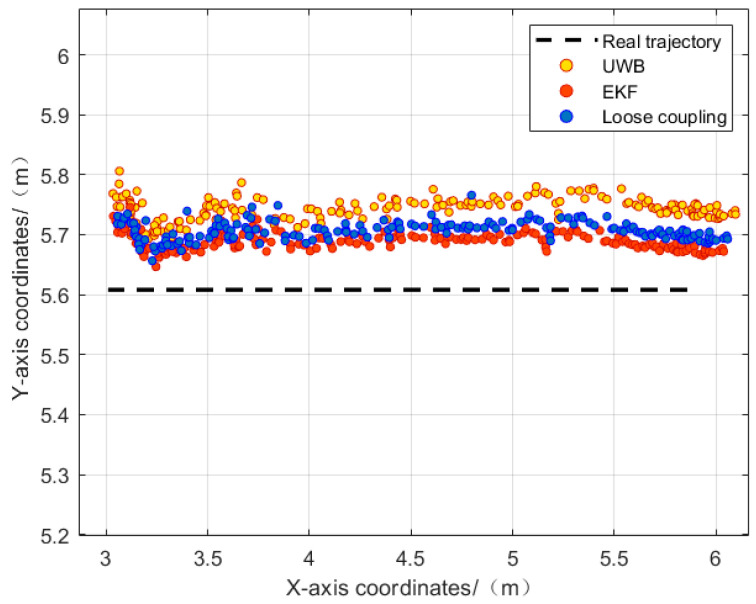
Comparison between solution trajectory and reference trajectory in the non-line-of-sight state.

**Figure 12 sensors-24-01710-f012:**
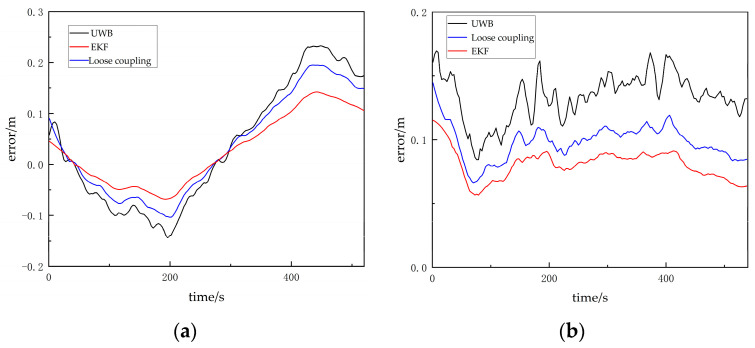
(**a**) Error in the x direction in non-line-of-sight scenarios; (**b**) error in the y direction in line-of-sight scenarios.

**Table 1 sensors-24-01710-t001:** Comparison of errors before and after UWB correction.

Direction	Index	Before Correction	Revised
X-axis	maximum	0.118	0.105
	average value	0.062	0.045
	standard deviation	0.034	0.028
Y-axis	maximum	0.305	0.086
	average value	0.076	0.020
	standard deviation	0.034	0.024

**Table 2 sensors-24-01710-t002:** Error statistics of three positioning methods in the line-of-sight state.

Direction	Index	UWB	Loose Coupling	EKF
X-axis	MAX	0.164	0.105	0.057
	RMSE	0.078	0.039	0.024
Y-axis	MAX	0.089	0.052	0.033
	RMSE	0.057	0.032	0.020

**Table 3 sensors-24-01710-t003:** Error statistics of three positioning methods in the non-line-of-sight state.

Direction	Index	UWB	Loose Coupling	EKF
X-axis	MAX	0.232	0.195	0.146
	RMSE	0.128	0.106	0.076
Y-axis	MAX	0.168	0.148	0.122
	RMSE	0.135	0.086	0.082

## Data Availability

Data are contained within the article.

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
