# Peer review of "Research on a Visual/Ultra-Wideband Tightly Coupled Fusion Localization Algorithm"

_sensors, 2024, doi:10.3390/s24051710_

Round 1

Reviewer 1 Report

Comments and Suggestions for Authors

The manuscript conducts preliminary exploration using the fusion of vision and UWB. Several experiments were carried out. But Many problems should be addressed.

1. What is the state vector and measurement of the adopted fusion model? The description is not clear.

2. How to do tight coupling? This part is not reflected.

3. The manuscript lacks comparisons with the state-of-the-art, especially in the aspects of NLOS distance, fusion, tightly coupling, etc.

4. The Introduction section lacks important reviews of NLOS distance, fusion, and tightly coupling.

5. The experiment scene doesn't include the UWB NLOS condition. It seems that the description is not approximate.

6. The experimental results and analysis behave inconsistently with the sections abstract and conclusions

Comments on the Quality of English Language

None

Reviewer 2 Report

Comments and Suggestions for Authors

"The manuscript presents a combined version of SLAM/UWB linked with the localization approach. The fundamental component of this technique is an Extended Kalman Filter (EKF), which fuses data by using the displacement increment produced by binocular vision ORB-SLAM and the positioning coordinates computed by UWB positioning as measurement information. Overall, the study is good. However, two major points are missing to complete the contribution. First, the literature review section is mandatory, discussing the accuracy obtained by previous studies' methodologies and their limitations. Second, the comparison presented in the study is not sufficient; benchmarking with at least two literature methods is required. At last, different sensor technologies and location problems should be mentioned, referring to seamless work at Traffic sensor location problem: Three decades of research."

Comments on the Quality of English Language

Moderate editing of English language required.

Reviewer 3 Report

Comments and Suggestions for Authors

This paper presents a visual/UWB tightly coupled fusion localization algorithm, where EKF is used to fuse UWB and visual SLAM data, and the Sage-Husa estimator is used to handle the influence of measurement noise on state estimation. The paper involves the following issues:

Although the authors claim the contributions in Section, they are not fundamentally novel. The paper does not involve theoretical novelty.

Section 3.2 introduces the Kalman fitler, which is well-known, and thus it is redundant.

The system state equation described by Eq. (20) is linear, and the system measurement equation described by Eq. (23) is also linear. There is no point to use the extended Kalman filter (EKF)? In fact, the EKF algorithm described in Section 3.3 is still the linear Kalman filter.

EKF involves a linearisation process of a nonlinear system by the Taylor expansion and thus the Jacobian matrix should be appeared in the system state equation. The authors should elaborate how to get Eq. (19) via Taylor expansion and how to get the Jacobian matrix. F in (20) is wrong, as for EKF it should be the dynamic Jacobian matrix evaluated at the estimated state of previous time.

The proposed method only considers the disturbance of unknown measurement noise statistics on system state estimation, what about the distance of unknown process noise statistics?

The performance evaluation does not evaluate the performance of using the adaptive Sage-Husa estimator in the proposed method.

In addition to the tightly coupled integration mode, there is also the loosely coupled integration mode. The authors should explain why the tightly coupled mode is chosen in the paper.

The authors should also explain why EKF is chosen in the paper. What about the other advanced nonlinear filters such as UKF and CKF?

The literature survey is incomplete.

a) The authors need to introduce the recent advances on handling the influence of measurement noise or abnormal measurement on system state estimation (see the following references for examples)

* Mahalanobis distance

Robust cubature Kalman filter with abnormal observations identification using Mahalanobis distance criterion for vehicular INS/GNSS integration, Sensors, 2019.

A novel adaptively-robust strategy based on the Mahalanobis distance for GPS/INS integrated navigation systems, Sensors, 2018.

Line and circle finding by the weighted Mahalanobis distance transform and extended Kalman filtering, 1994 IEEE International Symposium on Industrial Electronics, 1994.

* Innovation orthogonality

Robust unscented Kalman filtering with measurement error detection for tightly coupled INS/GNSS integration in hypersonic vehicle navigation, IEEE Access, 2019.

GPS/SINS positioning method based on robust UKF, 2012 International Conference on Industrial Control and Electronics Engineering, 2012.

Gain constrained robust UKF for nonlinear systems with parameter uncertainties, 2016 European Control Conference, 2016.

* Fault detection

A hypothesis test-constrained robust Kalman filter for INS/GNSS integration with abnormal measurement, IEEE Transactions on Vehicular Technology, 2023.

Strong tracking UKF method and its application in fault identification, Chinese Journal of Scientific Instrument 2008.

An enhanced MEMS-INS/GNSS integrated system with fault detection and exclusion capability for land vehicle navigation in urban areas, GPS Solutions, 2014.

* Maximum likelihood estimation of measurement noise covariance

Maximum Likelihood-based measurement noise covariance estimation using sequential quadratic programming for cubature Kalman filter applied in INS/BDS integration, Mathematical Problems in Engineering, 2021.

Identification of chemical processes with irregular output sampling, Control Engineering Practice, 2006.

Time series analysis and its applications: with R examples, New York: Springer, 2010.

b) The recent advances on improving EKF should also be introduced (see the following work as example).

Set-membership based hybrid Kalman filter for nonlinear state estimation under system uncertainty, Sensors, 2020.

A tighter set-membership filter for some nonlinear dynamic systems, IEEE Access, 2018.

AESMF based sensor fault diagnosis for RUAVs, 2012 24th Chinese Control and Decision Conference, 2012.

The paper needs to go through an English check. For example, in the sentence above Eq. (2), “as follows” should be “as follow”.

Comments on the Quality of English Language

The paper needs to go through an English check. For example, in the sentence above Eq. (2), “as follows” should be “as follow”.

Round 2

Reviewer 1 Report

Comments and Suggestions for Authors

I compare the revised manuscript and the old version carefully. The readability is very poor and it is impossible to know what has been changed. And I read through the simple and careless response. Both the modification and response are not satisfactory with me. In particular, the author deliberately omitted a very important question. There are mistakes in the revised version, such as the state vector. I still adhere to the six questions raised before, and hope that AE could give the author another chance to carefully revise it. I suggest the introduction section could cite more references. The following three references are suggested.

(1) Wital: A cots wifi devices based vital signs monitoring system using nlos sensing model

(2)LOS compensation and trusted NLOS recognition assisted WiFi RTT indoor positioning algorithm

(3)WiFi CSI-Based Long-Range Through-Wall Human Activity Recognition with the ESP32

Reviewer 2 Report

Comments and Suggestions for Authors

The authors try to address most of the comments however they did not improve the literature review section. I still need to see these two studies:  Traffic sensor location problem: Three decades of research and Deep Learning for Integrated Origin–Destination Estimation and Traffic Sensor Location Problems.

Comments on the Quality of English Language

None.

Reviewer 3 Report

Comments and Suggestions for Authors

First, the authors need to clearly refer to the places where the corresponding revisions are made in the responses, as only providing the page numbers for the revision places in the responses is not clear at all.

Regarding my comment 1, I do not see any revisions on pages 7-8 are related to the Jacobian matrix. Basically, the dynamic system is still linear and the measurement equation Eq. (19) is still linear. It is not clear how to get the measurement equation. If the state vector X_k and the measurement matrix H is plugged in (19), how to get (18) with dx_k and dy_k?

Regarding my comment 5, some key studies related to this paper are still missing.

a) The authors need to introduce the recent advances on handling the influence of measurement noise or abnormal measurement on system state estimation (see the following references for examples)

* Mahalanobis distance

Robust cubature Kalman filter with abnormal observations identification using Mahalanobis distance criterion for vehicular INS/GNSS integration, Sensors, 2019.

A novel adaptively-robust strategy based on the Mahalanobis distance for GPS/INS integrated navigation systems, Sensors, 2018.

Line and circle finding by the weighted Mahalanobis distance transform and extended Kalman filtering, 1994 IEEE International Symposium on Industrial Electronics, 1994.

* Innovation orthogonality

Robust unscented Kalman filtering with measurement error detection for tightly coupled INS/GNSS integration in hypersonic vehicle navigation, IEEE Access, 2019.

GPS/SINS positioning method based on robust UKF, 2012 International Conference on Industrial Control and Electronics Engineering, 2012.

Gain constrained robust UKF for nonlinear systems with parameter uncertainties, 2016 European Control Conference, 2016.

* Fault detection

A hypothesis test-constrained robust Kalman filter for INS/GNSS integration with abnormal measurement, IEEE Transactions on Vehicular Technology, 2023.

Strong tracking UKF method and its application in fault identification, Chinese Journal of Scientific Instrument 2008.

An enhanced MEMS-INS/GNSS integrated system with fault detection and exclusion capability for land vehicle navigation in urban areas, GPS Solutions, 2014.

* Maximum likelihood estimation of measurement noise covariance

Maximum Likelihood-based measurement noise covariance estimation using sequential quadratic programming for cubature Kalman filter applied in INS/BDS integration, Mathematical Problems in Engineering, 2021.

Identification of chemical processes with irregular output sampling, Control Engineering Practice, 2006.

Time series analysis and its applications: with R examples, New York: Springer, 2010.

b) The recent advances on improving EKF should also be introduced (see the following work as example).

Set-membership based hybrid Kalman filter for nonlinear state estimation under system uncertainty, Sensors, 2020.

A tighter set-membership filter for some nonlinear dynamic systems, IEEE Access, 2018.

AESMF based sensor fault diagnosis for RUAVs, 2012 24th Chinese Control and Decision Conference, 2012.

Round 3

Reviewer 1 Report

Comments and Suggestions for Authors

None

Comments on the Quality of English Language

None

Reviewer 3 Report

Comments and Suggestions for Authors

Most of my comments have been addressed in this round of revision. However, the response to my comment 5 on the literature survey still needs to improve.
1) The reference on fault detection is very old, while the latest advances on fault detection for handling measurement outliers are missing (see below).

Double-channel sequential probability ratio test for failure detection in multi-sensor integrated systems, IEEE Transactions on Instrumentation and Measurement, 2021.

2) The important technique of covariance matching for handling unknown or incorrect measurement noise covariance (see below) is missing in the paper.

Covariance matching based adaptive unscented Kalman filter for direct filtering in INS/GNSS integration, Acta Astronautica.

Comments on the Quality of English Language

The paper still involves grammar error, such as "errors. [32] A" in Para 1, Section 3.4.
